# The amount of Nck rather than N-WASP correlates with the rate of actin-based motility of Vaccinia virus

Angika Basant,[1] Michael Way[1,2]

**ABSTRACT** Vaccinia virus exiting from host cells activates Src/Abl kinases to phosphorylate A36, an integral membrane viral protein. Phosphorylated A36 binds the adaptors Nck and Grb2, which recruit N-WASP to activate Arp2/3-driven actin polymerization to promote viral spread. A36 also recruits intersectin, which enhances actin polymerization via AP-2/clathrin and Cdc42. How many viral and host molecules does such a virus-hijacked signaling network engage? To advance our quantitative understanding of this model signaling network, we determined the absolute numbers of the key molecules using fluorescent molecule-counting approaches in live cells. There are 1,156 ± 120 A36 molecules on virus particles inducing actin polymerization in HeLa cells. This number, however, is over 2,000 in mouse embryonic fibroblasts (MEFs), suggesting that A36 levels on the virion are not fixed. In MEFs, viruses recruit 1,032 ± 200 Nck and 434 ± 10 N-WASP molecules, suggesting a ratio of 4:2:1 for the A36:Nck:N-WASP signaling network. Loss of A36 binding to either secondary factor Grb2 or intersectin results in a 1.3- and 2.5-fold reduction in Nck, respectively. Curiously, despite recruiting comparable numbers of the Arp2/3 activator, N-WASP (245 ± 26 and 276 ± 66), these mutant viruses move at different speeds that inversely correlate with the number of Nck molecules. Our analysis has uncovered two unexpected new aspects of Vaccinia virus egress, numbers of the viral protein A36 can vary in the virion membrane and the rate of virus movement depends on the adaptor protein Nck.

**IMPORTANCE** Vaccinia virus is a large double-stranded DNA virus and a close relative of Mpox and Variola virus, the causative agent of smallpox. During infection, Vaccinia hijacks its host's transport systems and promotes its spread into neighboring cells by recruiting a signaling network that stimulates actin polymerization. Over the years, Vaccinia has provided a powerful model to understand how signaling networks regulate actin polymerization. Nevertheless, we still lack important quantitative information about the system, including the precise number of viral and host molecules required to induce actin polymerization. Using quantitative fluorescence microscopy techniques, we have determined the number of viral and host signaling proteins accumulating on virions during their egress. Our analysis has uncovered two unexpected new aspects of this process: the number of viral proteins in the virion is not fixed and the velocity of virus movement depends on the level of a single adaptor within the signaling network.

**KEYWORDS** Vaccinia virus, quantitative imaging, signaling networks, Nck, N-WASP, actin-based motility

Vaccinia virus is the most studied member of the poxvirus family and is best known for being used as the vaccine to protect against smallpox. It is a large double-stranded DNA virus, which undergoes a complex replication cycle in perinuclear cytoplasmic viral factories in infected cells (1). Replication results in the formation of infectious intracellular mature virions (IMV), which undergo kinesin-1-driven microtubule transport

Address correspondence to Angika Basant, angika.basant@crick.ac.uk, or Michael Way, michael.way@crick.ac.uk.

The authors declare no conflict of interest.

See the funding table on p. 17.

to disperse throughout the cell before being released when the infected host lyses (2–4). Prior to cell lysis, some IMV are also capable of becoming enveloped by a Golgi cisterna to form intracellular enveloped virions (IEVs) (1, 5, 6). This envelopment process, which is still not fully understood, is dependent on several viral proteins including the integral membrane protein B5 and the lipid-anchored F13 (7–10). Once formed, IEV undergo microtubule-based transport from their perinuclear site of assembly to the plasma membrane (11–14), where they fuse their outermost membrane to release extracellular enveloped virions (EEV) (1, 9). EEV are thought to promote the long-range spread of infection, but some virions can remain attached to the outer surface of a host cell after fusion. Known as cell-associated extracellular virions (CEV), these viruses can induce Arp2/3 complex-dependent actin polymerization beneath the virion (Fig. 1) (12, 15–17). The power of actin polymerization subsequently drives the CEV away from the host cell to enhance its spread into non-infected neighboring cells (15, 16, 18, 19).

The ability of IEV to move on microtubules as well as the stimulation of actin polymerization by CEV is dependent on A36, an integral viral membrane protein present in the Golgi that envelopes IMV during IEV formation (13, 14, 17, 20–22). In contrast to B5 and other integral viral membrane proteins in the Golgi that wrap IMV, A36 does not end up in EEV and is only retained in the plasma membrane beneath CEV (Fig. 1) (9, 23). A36 facilitates microtubule transport of IEV by recruiting kinesin-1 to the virus (13). It achieves this by interacting directly with the tetratricopeptide repeats of the kinesin

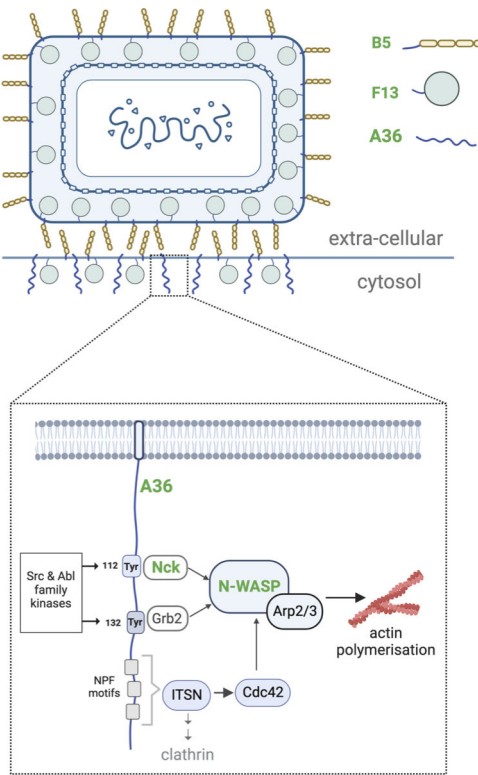

**FIG 1** CEV and the signaling network that induces actin polymerization. A schematic showing the localization of the Vaccinia virus proteins A36, B5, and F13 in CEV on the plasma membrane. A36 and B5 are integral membrane proteins, while F13 associates with the membrane via lipidation. Inset shows the signaling network recruited by the cytoplasmic tail of A36 inducing actin polymerization. Tyr112 and Tyr132 in A36, when phosphorylated by Src and Abl family kinases, recruit the adaptor proteins Nck and Grb2. Nck and Grb2 interact with N-WASP, which results in the activation of the Arp2/3 complex and stimulation of actin polymerization. NPF motifs in the C-terminus of A36 recruit the RhoGEF intersectin, which recruits AP2/clathrin and also activates Cdc42 so it can bind N-WASP. While Nck is essential for actin tail formation, Grb2 and Cdc42 are not. Molecules labeled in green have been quantitatively analyzed in this study.

light chain via a bipartite tryptophan acidic motif that is also found in cellular kinesin-1 binding proteins (24–26). After IEV fuse with the plasma membrane, CEV attached to the outside of the cell locally activate Src and Abl family kinases to induce the phosphorylation of tyrosine 112 and 132 in the cytosolic tail of A36 (Fig. 1) (17, 27–29). Phosphorylation of tyrosine 112 and 132 results in the recruitment of the adaptors Nck and Grb2 via their SH2 domains, respectively (17, 30). Nck subsequently recruits N-WASP via WIP to stimulate the actin-nucleating activity of the Arp2/3 complex beneath CEV (31–33). The resulting actin polymerization drives CEV motility, enhancing cell-to-cell spread of infection (14). Not surprisingly, the amount of phosphorylation-competent A36 influences the length of actin tails as well as the speed of virus movement (34). Grb2, in contrast to Nck, is not essential for this process, but its recruitment stabilizes the signaling complex, making actin assembly more robust (30, 35). In addition, three NPF motifs near the C-terminus of A36 interact with the RhoGEF intersectin to recruit Cdc42 and clathrin, further enhancing actin polymerization beneath CEV (Fig. 1) (34, 36, 37).

We know the identity of the viral and key host components as well as their interactions and roles in CEV-stimulated actin assembly. The turnover rates of Nck, Grb2, WIP, and N-WASP beneath CEV have also been determined (35). However, to obtain a full quantitative understanding of virus-induced actin polymerization, we need to establish the number of molecules participating in this signaling network. Such an investigation will also allow comparative analyses with other networks in cell physiology to uncover basic signaling principles. Taking advantage of fluorescent nanocages with defined numbers of GFP molecules (38), we recently determined that IEV recruit 320 kinesin-1 motor complexes (4). Using a similar approach, we have now determined the number of A36 molecules beneath CEV together with B5 and F13, which are essential for IEV assembly, as well as Nck and N-WASP that are required for CEV-induced actin polymerization.

## RESULTS

### Quantification of the number of A36 molecules associated with CEV

To measure the number of A36 molecules on virus-inducing actin tails using quantitative fluorescent imaging, we first generated a recombinant virus expressing an A36-TagGFP2 fusion from its genomic locus together with RFP-A3 to mark viral cores (Fig. S1A). The A36-TagGFP2 recombinant virus formed plaques that were slightly smaller in diameter than Western Reserve (WR), the parental strain (1.63 ± 0.12 vs 1.95 ± 0.07 mm) (Fig. S1B). Importantly, A36-TagGFP2 was incorporated in late-stage virus particles, which localized to the tip of actin tails (Fig. 2A). The intensity of A36-TagGFP2 was substantially higher than membrane-tethered nanocages we have previously used to generate a fluorescence calibration curve in live cells [Fig. 2B; (4)]. The upper limit of our calibration curve is 180 molecules [Fig. S3A; (4)]. To examine whether the curve we generated would be valid for higher fluorescence intensities, we acquired images of 60mer and 120mer nanocages at 4× and 3× exposure times, respectively. The average measurements obtained from these experiments were consistent with the expected values for 240mer and 360mer nanocages (Fig. 2C). To quantitively determine the number of A36 molecules on virus particles inducing actin tails, we imaged and analyzed confocal Z-stacks of HeLa cells infected with the A36-TagGFP2 virus from 8 hours post-infection, using LifeAct-iRFP and RFP-A3 as actin and viral markers, respectively (Fig. 2D). Co-localization of A36-TagGFP2 and RFP-A3 signals occurs on IEV and CEV, but the inclusion of an actin marker ensures that only CEV are analyzed, as actin tails are not generated by IEV in the cytoplasm of infected cells (12, 13). This analysis revealed that CEV on the tips of actin tails have 1,156 ± 120 A36 molecules. As this value is much higher than our experimentally established calibration curve, we also measured A36-TagGFP2 intensities on actin tails at 0.5× and 0.25× exposure times (Fig. 2D). The average intensity values obtained from these shorter exposure times lie along the extrapolated calibration curve, suggesting that our determination of A36 numbers is reliable.

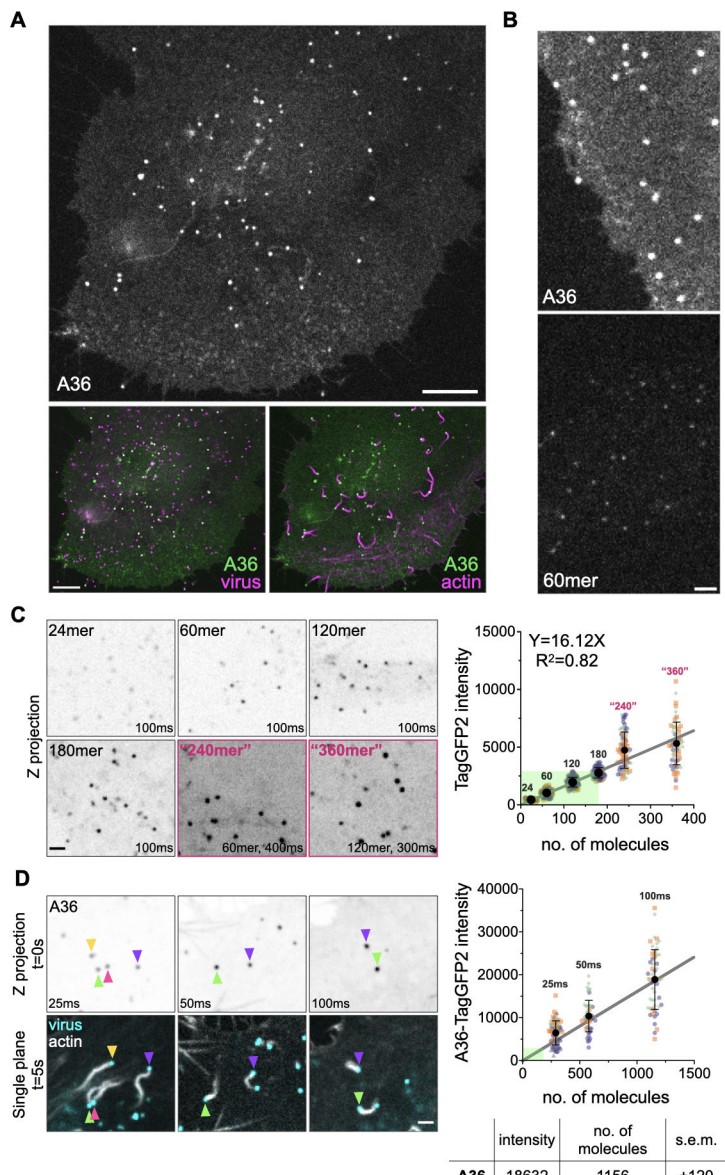

**FIG 2** Determining the number of A36 molecules on CEV inducing actin polymerization. (A) Representative single-plane spinning disc confocal image of a live HeLa cell stably expressing LifeAct-iRFP and infected with a recombinant virus expressing A36-TagGFP2 and RFP-A3 at their endogenous loci. The image is acquired at 9 hours post-infection. Scale bars = 10 µm. (B) Representative single-plane confocal images comparing fluorescence intensity of A36-TagGFP2 on virus particles with TagGFP2-labeled 60mer nanocages in live HeLa cells. Scale bar = 3 µm. (C) Representative inverted average-intensity Z-projections of TagGFP2-tagged nanocages transiently expressed in live HeLa cells treated with 500 nM AP21967. Images were acquired at 100 ms exposure. 60mers were imaged at 4× exposure to generate "240mer" intensities, and 120mers were imaged at 3× exposure to generate "360mer" intensities. Scale bar = 2 µm. The graph shows the quantification of background-subtracted raw integrated fluorescence intensities of nanocages. Linear line of regression is fitted to data from 24-, 60-, 120-, and 180mers. A total of 47–110 membrane-tethered nanocages were measured per condition over three independent experiments. A detailed version of the experimental data used to generate the calibration curve (green shaded area) is shown in Fig. S3A. (D) Representative inverted average-intensity Z-projections of A36-TagGFP2 at virus particles generating actin tails in live HeLa cells. Images were acquired at indicated exposure times. Colored arrowheads indicate individual virus particles. Single-plane images in the lower panel show the positions of each particle 5 seconds after the Z-stack acquisition. Actin is labeled with LifeAct-iRFP, and all virus particles are labeled with RFP-A3. Scale bar = 2 µm. The graph shows the quantification

**FIG 2** (Continued)

of background-subtracted raw integrated A36-TagGFP2 intensities at actin-polymerizing virus particles, overlaid on the calibration curve generated in panel C. A total of 50–57 virions were measured per condition over three independent experiments. All error bars represent SD and the distribution of data from each experiment is shown using a "SuperPlot." The table shows the calculated number of A36 molecules on CEV-inducing actin tails in HeLa cells (s.e.m. = standard error of the mean).

## F13 is more abundant than B5 and A33 on CEV

In addition to A36, the outer membrane of Vaccinia has additional viral proteins, A33, A34, A56, E2, F12, B5, and F13, the latter two of which are required for the formation of IEV, the precursor to CEV (9, 39, 40). B5, like A36, is an integral membrane protein, while F13 associates with the viral membrane via lipidation. Given the essential role of B5 and F13 in the assembly of IEV, the precursor to CEV, we investigated how the abundance of B5 and F13 compares to A36. We, therefore, generated recombinant viruses expressing B5-TagGFP2 or TagGFP2-F13 at their endogenous loci in a parental virus expressing RFP-A3 to determine the numbers of B5 and F13 molecules associated with CEV nucleating actin tails (Fig. S1A). As is the case with A36-TagGFP2, recombinant viruses expressing B5-TagGFP2 and TagGFP2-F13 formed plaques that were slightly smaller than the parental WR strain (Fig. S1). The overall expression levels of the tagged proteins in infected cells are also reduced when compared to untagged controls (Fig. S1). B5-TagGFP2 and TagGFP2-F13 positive virus particles were observed at the tips of actin tails in infected cells, with TagGFP2-F13 appearing substantially brighter than B5-TagGFP2 (Fig. 3A). To quantify their absolute numbers, we imaged and analyzed Z sections of infected cells as described for A36-TagGFP2. Based on the fluorescence intensities obtained with our calibration curve, we calculated that CEV particles in HeLa cells inducing actin tails have 3,107 ± 602 and 8,729 ± 688 B5 and F13 molecules, respectively (Fig. 3B and C). These numbers are considerably higher than that of A36; however, B5 and F13, unlike A36, are present in the plasma membrane beneath CEV as well as the virion membrane (Fig. 1). We believe the values for F13 and B5, though beyond our calibration curve, are valid as their signal was collected in the linear response range of the camera (absence of saturation). We compared these results with the levels of A33, another transmembrane viral protein present in both membranes, which interacts with B5 (41, 42). We generated a recombinant virus expressing TagGFP2-A33 at its endogenous locus in an RFP-A3 background (Fig. S2A and B). Quantitative analysis reveals that the number of A33 molecules (3,411 ± 429) present on CEV inducing actin tails in HeLa cells is comparable to that of B5 (3,107 ± 602) (Fig. S2C).

## What is the ratio of A36:Nck:N-WASP on virions nucleating actin?

Once phosphorylated, A36 recruits a signaling network involving the adaptor Nck and N-WASP to activate the Arp2/3 complex to induce an actin tail. Nck and N-WASP are essential for this process, but how many molecules of each are recruited beneath CEV on the plasma membrane? To address this question, we generated cell lines stably expressing TagGFP2-Nck1 in MEFs lacking Nck1 and Nck2, as well as TagGFP2-N-WASP in N-WASP null MEFs (Fig. S4). Using these cells ensures that all the Nck or N-WASP molecules are fluorescently labeled. However, these are a different host cell type from HeLa, which were used to determine the number of A36 molecules associated with CEV.

Virion composition and structure are not expected to vary depending on the host cell type, but this has never formally been investigated. To determine whether A36 numbers are comparable between MEF and HeLa cells, we quantified A36-TagGFP2 molecules on virions inducing actin tails in Nck+/+ and N-WASP+/+ parental MEFs. Surprisingly, we find that the number of A36 molecules in both MEF cell lines is considerably higher than in HeLa cells (Fig. 4A). We confirmed that nanocage intensities were comparable between these cell types (Fig. S3B), suggesting that A36 incorporation into CEV is indeed

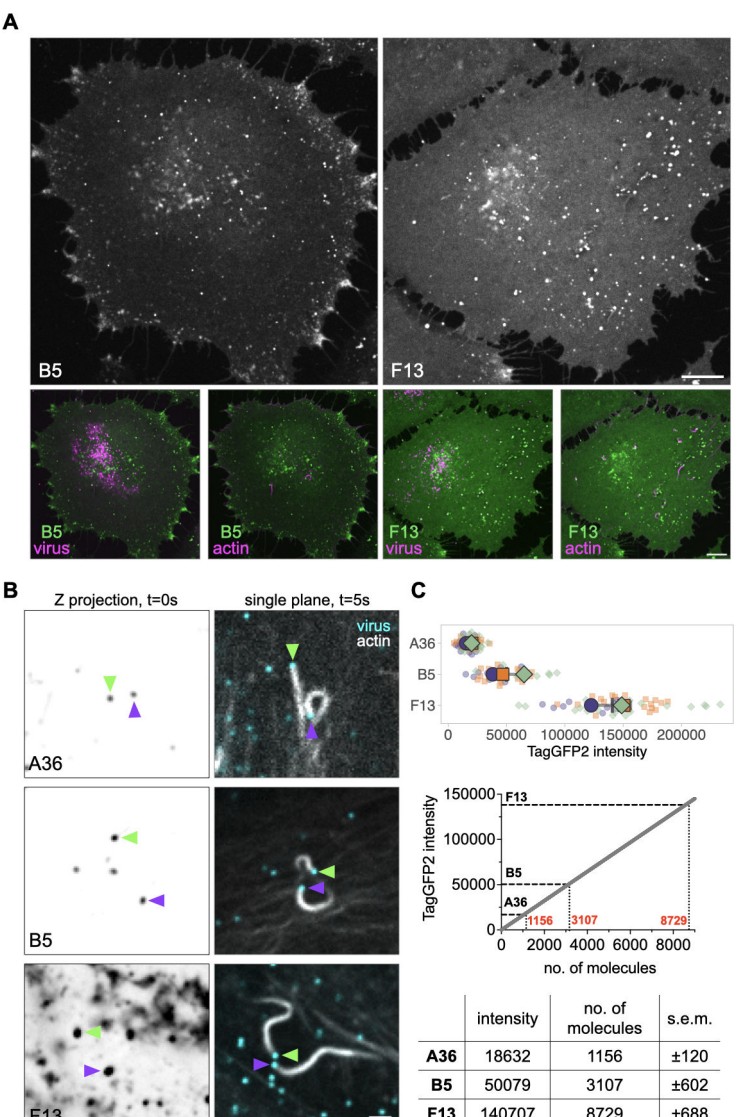

**FIG 3** Determining the number of B5 and F13 molecules on CEV inducing actin tails. (A) Representative single-plane confocal images of live HeLa cells infected with a recombinant virus expressing B5-TagGFP2 or TagGFP2-F13 at their endogenous loci. Actin is labeled with LifeAct-iRFP, and all virus particles are labeled with RFP-A3. Images are acquired at 9 hours post-infection. Scale bar = 10 µm. (B) Representative inverted average-intensity Z-projections of A36-TagGFP2, B5-TagGFP2, and TagGFP2-F13 at virus particles generating actin tails in live HeLa cells. Colored arrowheads indicate individual virus particles. Single-plane images in the right panel show the positions of each particle 5 seconds after the Z-stack acquisition. Actin is labeled with LifeAct-iRFP, and all virus particles are labeled with RFP-A3. Scale bar = 2 µm. (C) The top graph shows the quantification of background-subtracted raw integrated TagGFP2 intensities at actin-polymerizing virions. All error bars represent SD, and the distribution of data from each experiment is shown using a "SuperPlot." A total of 39–54 particles were measured per recombinant virus over three independent experiments. The lower graph shows average measurements for each molecule superimposed on the calibration curve generated in Fig. 2C. The table shows the calculated number of A36, B5, and F13 molecules at actin-polymerizing virus particles in HeLa cells (s.e.m. = standard error of the mean).

higher in MEFs. We wondered whether this difference relates to the cell type and/or host species. To test whether this is the case, we determined A36 numbers on virus inducing actin polymerization in human lung epithelial A549 cells. We found that CEV-forming

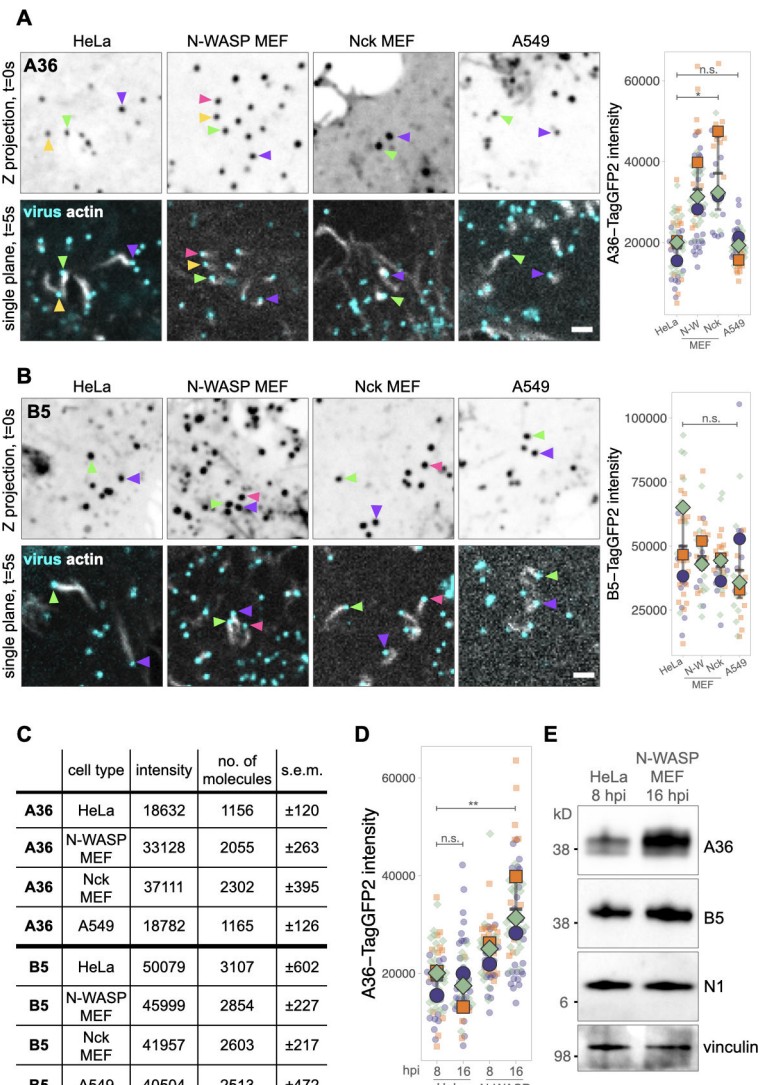

**FIG 4** Comparing the numbers of A36 and B5 at CEV across host cell types. (A and B) Representative inverted average-intensity Z-projections of A36-TagGFP2 (A) and B5-TagGFP2 (B) at virus particles generating actin tails in the indicated cell lines. Colored arrowheads indicate individual virus particles. Single-plane images in the lower panels show the positions of these particles 5 seconds after the Z-stack acquisition. Actin is labeled with LifeAct-iRFP (stably expressed in HeLa and transiently expressed in other cell types). All virus particles are labeled with RFP-A3. Scale bar = 2 µm. The graphs show the quantification of background-subtracted raw integrated TagGFP2 intensities at actin-polymerizing virus particles. All error bars represent SD, and the distribution of data from each experiment is shown using a "SuperPlot." A total of 29–62 (A36) or 26–41 (B5) particles were measured per condition over three independent experiments. Tukey's multiple comparison test was used to determine statistical significance; ns, not significant; $P > 0.05$; and $*P \leq 0.05$. (C) The table shows the calculated number of A36 and B5 molecules at virions inducing actin tails in the different host cell lines. (D). Graph showing the quantification of background-subtracted raw integrated A36-TagGFP2 intensities at actin-polymerizing virus particles at 8 and 16 hours post-infection (hpi) in HeLa and N-WASP MEF host cells. All error bars represent SD, and the distribution of data from each experiment is shown using a "SuperPlot." A total of 48–62 particles were measured per condition over three independent experiments. Tukey's multiple comparison test was used to determine statistical significance; ns; $P > 0.05$; and $**P \leq 0.01$. (E) Immunoblot analyses of total cell lysates from WR-infected HeLa and N-WASP MEF cells at 8 and 16 hpi, respectively. N1 and vinculin represent viral and host protein loading controls.

actin tails in A549 cells have comparable numbers of A36 molecules as we measured in HeLa (A549 1,165 ± 126 vs HeLa 1,156 ± 120). We next sought to determine if these host-dependent differences are also true for B5, another integral membrane Vaccinia protein. Measuring the intensity of B5-TagGFP2 on CEV in HeLa, A549, and the two MEF cell lines reveals that, in contrast to A36, the numbers of B5 molecules are comparable between the four different cell lines (Fig. 4B).

As the Vaccinia lifecycle appears to proceed faster in HeLa cells than MEFs, our lab typically analyzes actin tails at 8 and 16 hours post-infection in these cell types, respectively (35). To investigate whether A36 levels on CEV depend on time post-infection, we compared A36-TagGFP2 intensities in HeLa and MEF cells at 8 and 16 hours post-infection. We find that A36 numbers on CEV particles do not vary with time in HeLa cells. In contrast, in MEFs, the level of A36 associated with CEV increases as the infection progresses, but even at 8 hours post-infection, more A36 molecules are present than in HeLa (Fig. 4D). The reason for this difference is not immediately clear although it is striking that the bulk levels of A36, but not B5, are considerably higher in N-WASP+/+ parental MEFs than HeLa cells (Fig. 4E).

Having determined the number of A36 molecules on CEV in MEF cells, we measured the number of Nck and N-WASP molecules associated with the virus. TagGFP2-tagged Nck and N-WASP were both recruited beneath CEV-inducing actin tails (Fig. 5A). Quantification of the fluorescence intensity of the TagGFP2 signals in Z stacks indicates that each CEV recruits on average 1,032 ± 200 and 434 ± 10 Nck and N-WASP molecules, respectively (Fig. 5C and D). Taken together with our data on A36 numbers on actin tail forming particles in MEFs in Fig. 4C, this suggests that A36, Nck, and N-WASP are present in a ~4:2:1 ratio. This ratio is our best estimate, as unfortunately due to the nature of the fluorescent tag and lack of double Nck/N-WASP null cells, we cannot simultaneously measure the numbers of any pair of the three molecules in a single cell.

## The level of Nck and not N-WASP correlates with the rate of CEV motility

While Nck and N-WASP are essential for CEV-induced actin polymerization, the virus also recruits Grb2 and intersectin, which act as secondary factors to enhance actin tail formation (Fig. 1). Our previous FRAP (fluorescence recovery after photobleaching) analysis demonstrates that Grb2 helps stabilize Nck and N-WASP beneath CEV during virus-induced actin polymerization (35). To determine whether these factors also impact the absolute number of Nck and N-WASP molecules, we infected cells with either the A36-Y132F or A36ΔNPF mutant viruses recruited to CEV. Our measurements show that mutation of Tyr132 to phenylalanine (loss of Grb2) reduced the level of Nck and N-WASP by ~24% and 44%, respectively(Fig. 6A and B). However, in the absence of the A36 NPF motifs, Nck numbers were ~61% lower while N-WASP was reduced by ~36% (Fig. 6A and B). Interestingly, for both the A36-Y132F and A36ΔNPF mutant viruses, the number of N-WASP molecules was reduced to similar levels (245 ± 26 and 276 ± 66 respectively).

Nck and N-WASP association with CEV generates Arp2/3-mediated actin polymerization that drives the motility of the virus. As N-WASP directly activates the Arp2/3 complex, we would predict that N-WASP numbers alone ultimately determine the speed of the virion. To examine if this is indeed the case, we measured the speed of wild-type, A36-Y132F, and A36ΔNPF viruses in HeLa cells stably expressing LifeAct-iRFP. Surprisingly, despite having similar levels of N-WASP, the A36-Y132F and A36ΔNPF viruses move at different speeds (Fig. 6C). The A36-Y132F virus is slightly faster than the wild-type WR strain (0.2 ± 0.01 compared to 0.17 ± 0.01 µm/second), but the speed of A36ΔNPF virus is significantly increased (0.24 ± 0.02 µm/second). To test whether these differences are caused by varying levels of A36 associated with CEV, we generated recombinants expressing A36(132F)-TagGFP2 and A36ΔNPF-TagGFP2 at the endogenous locus. The numbers of mutant A36 molecules present at CEV generating actin tails were comparable to wildtype [Fig. 6C, A36-TagGFP2, 1,186 ± 256; A36(132F)-TagGFP2, 1,242 ± 140; A36ΔNPF-TagGFP2, 966 ± 185]. Our quantitative observations suggest that virus speed is inversely proportional to the number of Nck molecules recruited by the virus.

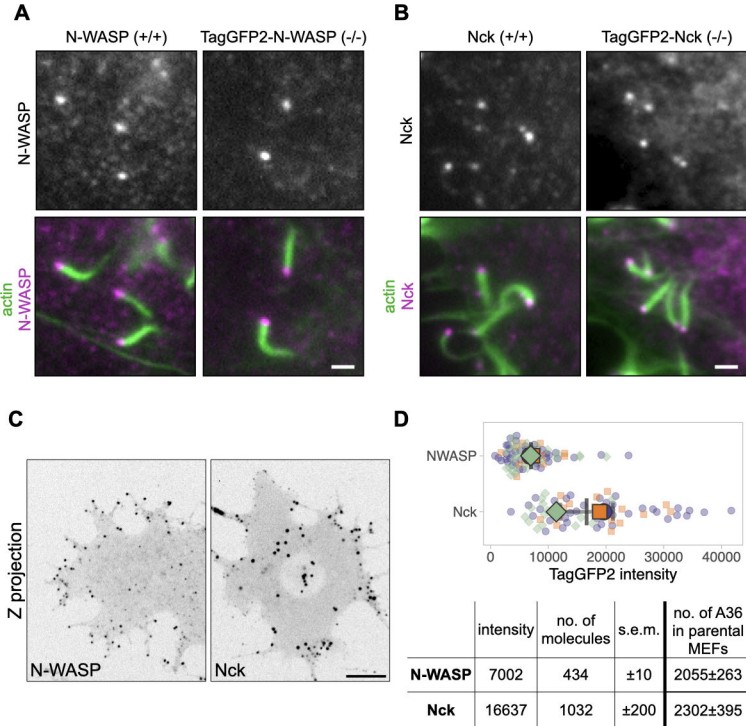

**FIG 5** Determining the number of Nck and N-WASP molecules on CEV. (A) Representative immuno-fluorescence images showing the recruitment of N-WASP to actin tails in Vaccinia-infected MEF cells expressing TagGFP2-N-WASP in an N-WASP-null background or parental wild-type cells at 16 hours post-infection. N-WASP was detected with an antibody, and actin was stained with phalloidin. Scale bar = 1 μm. (B) Representative immunofluorescence images showing the recruitment of Nck to actin tails in Vaccinia-infected MEF cells expressing TagGFP2-Nck in a Nck-null background or parental wild-type cells at 16 hours post-infection. Nck was detected with an antibody, and actin is stained with phalloidin. Scale bar = 1 μm. (C) Representative inverted average-intensity Z-projections of TagGFP2-Nck and TagGFP2-N-WASP in live Vaccinia WR-infected MEF cells stably expressing the corresponding tagged protein in Nck1/Nck2 −/− and N-WASP −/− backgrounds, respectively. Images are taken at 16 hours post-infection. Scale bar = 10 μm. (D) The graph shows the quantification of background-subtracted raw integrated TagGFP2 intensities at actin-polymerizing virus particles. All error bars represent SD, and the distribution of data from each experiment is shown using a "SuperPlot." A total of 71–85 virions were measured per condition over three independent experiments. The table shows the calculated number of Nck and N-WASP molecules at actin-polymerizing virus particles in MEF cells.

## DISCUSSION

Over the years, Vaccinia has provided a powerful model to understand the function and regulation of Nck:N-WASP signaling and its role in stimulating Arp2/3-driven actin polymerization. Our current analysis provides new quantitative information concerning the signaling network driving this process. We found that Nck is twice as abundant as N-WASP but half as much as A36, the viral protein ultimately initiating actin tail formation. Overall, there appears to be a 4:2:1 ratio for the A36:Nck:N-WASP signaling network in MEFs. These values are consistent with the 2:1 ratio for Nck:N-WASP determined from modeling and live cell imaging of artificially clustered Nck SH3 domain on the plasma membrane (43). Our analysis goes beyond ratios as it now provides absolute numbers for Nck and N-WASP as well as A36, the activator of the signaling network. This constitutes the first such characterization of a host-viral signaling network.

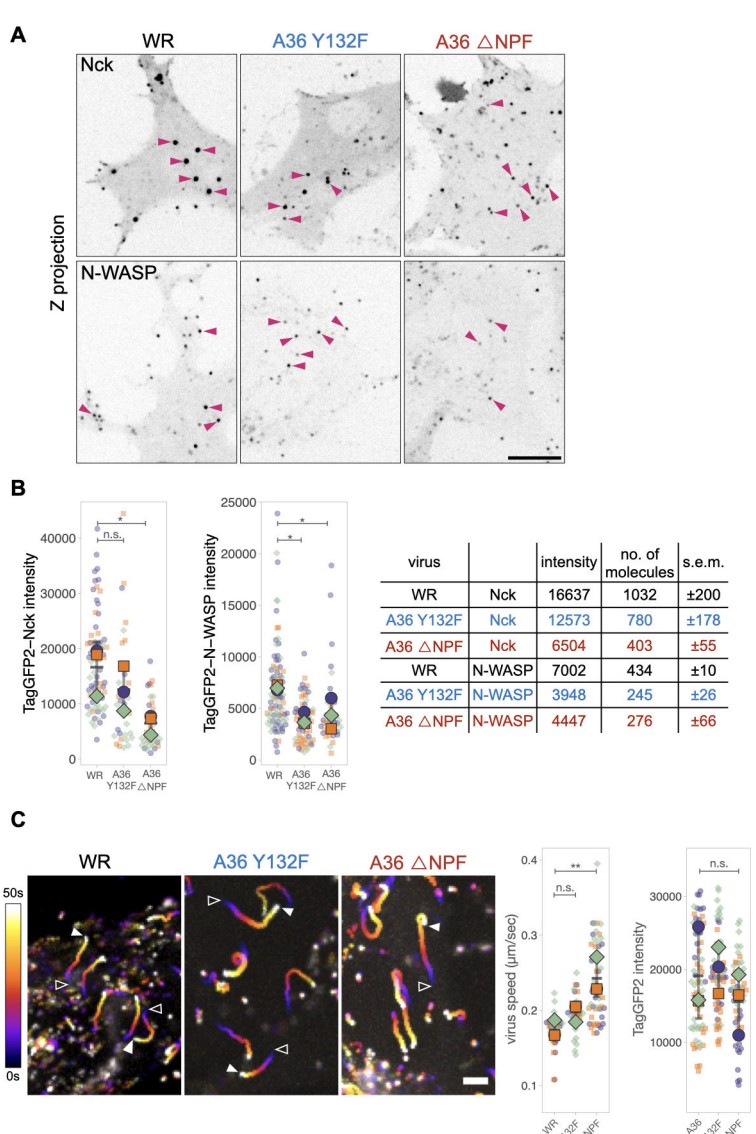

FIG 6   Nck and N-WASP are reduced on CEV in the absence of Grb2 and intersectin. (A) Representative inverted average-intensity Z-projections of TagGFP2-Nck and TagGFP2-N-WASP at virus particles generating actin tails in live MEF cells infected with Vaccinia Western Reserve virus or the indicated A36 mutants. Pink arrowheads indicate particles generating actin tails as discerned by a transiently expressed LifeAct-iRFP and RFP A3, which label viral cores. The LifeAct-iRFP and RFP-A3 channels are not shown for clarity. Scale bar = 10 µm. (B) Graphs showing the quantification of background-subtracted raw integrated intensities of TagGFP2-Nck and TagGFP2-N-WASP at actin-polymerizing A36 mutant virus particles. All error bars represent SD, and the distribution of data from each experiment is shown using a "SuperPlot." A total of 34–49 (Nck) or 41–85 (N-WASP) particles were measured per condition over three independent experiments. Dunnett's multiple comparison test was used to determine statistical significance; ns; $P > 0.05$; and $*P \leq 0.05$. The table shows the calculated number of Nck and N-WASP molecules at actin-polymerizing virus particles in the indicated A36 mutants. (C) Temporal color-coded representation of time-lapse movies tracking the motility of the indicated RFP-A3-labeled virus over 50 seconds in HeLa cells at 8 hours post-infection. Images were recorded every second, and the position of virus particles at frame 1 (open triangles) and frame 50 (white triangles) is indicated. Scale bar = 3 µm. The graph (left) shows the quantification of virus speed. A total of 31–51 tracks were analyzed per condition over three independent experiments. The graph on the right shows the quantification of background-subtracted raw integrated intensities of TagGFP2 at actin-polymerizing recombinant virus

**FIG 6** (Continued)

particles expressing either A36-TagGFP2, A36(132F)-TagGFP2, or A36ΔNPF-TagGFP2. A total of 59–67 particles were measured per condition over three independent experiments. All error bars represent SD, and the distribution of data from each experiment is shown using a "SuperPlot." Dunnett's multiple comparison test was used to determine statistical significance; ns; $P > 0.05$; and $**P ≤ 0.01$.

## Speed of the virus depends on the level of Nck and not N-WASP

Previous studies have taken advantage of the known structure of viruses to generate fluorescence calibration standards by tagging defined numbers of viral capsid components with fluorescent proteins (44–46). More recently, engineered nanocages with fixed numbers of GFP molecules have been used to provide quantitative insights into N-WASP and Arp2/3-driven endocytosis in yeast and human cells (38, 47). Using a similar imaging approach, we find that CEV-inducing actin polymerization in MEFs recruits just over 400 molecules of N-WASP. This value is considerably higher than the maximum N-WASP numbers measured for endocytic patches in yeast (budding yeast 102 and fission yeast 138) (47). Interestingly, however, in human cells, ~200 copies of the Arp2/3 complex are recruited during clathrin-mediated endocytic events (38). Previous biochemical and structural analyses demonstrate that N-WASP dimers are more efficient in activating Arp2/3 and that the complex contains two N-WASP-binding sites (48–50). This would suggest that an endocytic site in human cells will also recruit ~400 N-WASP molecules, which is similar to the values we have observed associated with CEV. This is surprising, as the area under a CEV is considerably larger than an invaginating endocytic pit, which is typically 60–100 nm in diameter in human cells (51, 52). Based on our recent measurements of Vaccinia virion dimensions in cryo-electron tomograms of infected cells, the area under a CEV, which corresponds to the outer membrane of IEV, is 405,037 $nm^2$ (53). This value is ~13× higher than the surface of an endocytic pit (based on a radius of 50 nm). This suggests that the density of the signaling network beneath Vaccinia CEV is significantly less than that of an endocytic patch.

Loss of Grb2 or intersectin recruitment leads to even lower levels of N-WASP (245 ± 26 and 276 ± 66, respectively) at CEV. However, the system does not collapse, and the rate of virus motility even increases. Moreover, while the recruited levels of N-WASP in the two mutants are similar, in the absence of intersectin (A36ΔNPF), the virus moves significantly faster than when Grb2 is absent (A36-Y132F). The striking difference between these two mutant viruses is the level of Nck recruited to the CEV, with the numbers being 61% lower for the A36ΔNPF mutant compared to the wild-type virus. Nck and N-WASP are both essential for Vaccinia-induced actin polymerization, but how their relative levels impact the rate of actin-based motility is not established. Previous analysis reveals the density of artificially clustered VCA, the N-WASP domain responsible for activating Arp2/3, does not affect downstream actin-based motility (54). Interestingly, the same group also found that increasing the density of artificially clustered Nck SH3 domains on the plasma membrane increases the motility of actin structures (43). In their model system, Nck is permanently attached to the plasma membrane and not free to exchange as under CEV. This may underlie the difference in the trends of our observations. However, together with these previous studies, our data indicate that Nck numbers and density are critical determinants of actin-based motility in cells. It will be interesting to investigate how these parameters influence the turnover of Nck and N-WASP beneath CEV (35).

Why do Nck numbers impact Vaccinia motility? N-WASP/WASP activates Arp2/3-mediated actin polymerization more efficiently when acting as a dimer (48–50, 55). Previous simulations based on results from *in vitro* experiments suggest that when a protein that would dimerize N-WASP is high, the probability of two N-WASP molecules engaging the same dimerizer is low (55). Namely, each dimerizer would have a single N-WASP, which would be less efficient at stimulating the actin-nucleating activity of the Arp2/3 complex. When the dimerizer concentration is optimal, it is more likely that two N-WASP

molecules will interact with the same dimerizer making actin nucleation more efficient. Our previous *in vitro* pulldown experiments demonstrated that two of the three SH3 domains in Nck were capable of binding to N-WASP suggesting that this adaptor can function as a dimerizer (31). Thus, it is possible that the increase in the velocity of the A36ΔNPF virus is due to the concentration of Nck being closer to the optimal level required to fully dimerize N-WASP, making actin nucleation more efficient. It remains to be established why the lack of intersectin recruitment by A36 affects the levels of Nck so profoundly. The loss of the A36 NPF motifs results in reduced clustering of A36 (34), which is expected to reduce the local density of Nck under CEV. However, it should not reduce the total number of Nck molecules. We, therefore, favor a hypothesis that the absence of intersectin results in the loss of an unknown Nck-stabilizing protein in addition to Cdc42 and AP2/clathrin.

## Implications of A36 density under CEV

A36 is present in the outer IEV membrane, which becomes incorporated into the plasma membrane below CEV during Vaccinia egress from the host cell. A36 contains a single helical transmembrane domain, which is estimated to be approximately 1.2 nm wide. Assuming that no membrane is lost, and proteins do not diffuse significantly when IEVs fuse with the plasma membrane, our data indicate that A36 is present at a density of 0.2–0.5 molecules per 100 $nm^2$ based on the outer surface of 405,037 $nm^2$ for IEV. Interestingly, it has been determined that 5,800 molecules of the actin-nucleating protein, ActA, of Listeria are present on the surface of the bacterium (56). The bacterial surface area is substantially larger than the membrane beneath CEV; however, the density of ActA molecules (0.09–0.3 molecules/100 $nm^2$) is in a similar range to that of A36 (56).

Unexpectedly, we found that the number of A36 molecules associated with CEV varies depending on the cell type. It can also be influenced by the length of the infection. This is not the case with B5, which appears to remain relatively constant. The reason for the variation in A36 numbers is not clear, but it might be related to its level of expression in the infected cell (Fig. 4E). Moreover, the variation in A36 numbers suggests that viral protein composition in membranes around IEV and subsequent forms of the virus may not be fixed for all viral proteins. Consistent with this, previous analysis reveals that approximately one-third of EEV lack A56, another integral IEV membrane protein (9, 57).

Previous biochemical pulldowns and immunoprecipitation approaches have demonstrated that there are multiple interactions between IEV membrane proteins including A33:B5 (41, 42, 58), A34:B5 (20, 59, 60), A34:A36 (20), A33:A36 (20, 61, 62), A36:F12 (63), and F12:E2 (39, 64). These multiple associations suggest that the viral proteins likely form an immobile lattice within the membrane. Our measurement of the number of A36, B5, and F13 molecules in CEV is an important step in fully determining the precise organization of proteins in the viral membrane. B5 and F13 are present in the membrane surrounding CEV as well as together with A36 in the plasma membrane beneath CEV. Assuming B5 and F13 are equally partitioned between the two membranes, then the levels of B5 and F13 are 1.3 and 3.8-fold higher than A36 beneath the CEV. It is striking that the levels of A36 and B5 are similar given there is no evidence that these proteins associate with each other (64). We also found that the levels of A33 are also similar to B5. Given its interactions, it will be interesting to measure the levels of A34. If the levels of A34, which is also an integral membrane protein, is comparable to A36, A33 and/or B5, it would suggest that the viral protein density in membranes around IEV is surprisingly low and that they do not form a rigid protein lattice that covers the whole virion. In addition, the absence of an immobile viral protein lattice may explain why clathrin is able to cluster A36 after IEV fuse with the plasma membrane (34).

In summary, our quantitative approach in determining precise numbers of molecules at CEV has unexpectedly revealed that A36 levels in the viral membrane are variable. It has also demonstrated that the velocity of virus movement depends on the adap-

tor protein Nck rather than N-WASP, which activates Arp2/3 complex-dependent actin polymerization.

## MATERIALS AND METHODS

### Expression constructs and targeting vectors

The expression vector pE/L-LifeAct-iRFP670 and the plasmid for the 180-mer nanocage were previously made in the Way lab (4, 65). Plasmids for 12-, 24-, 60-, and 120-mer TagGFP2 nanocage expression were obtained from the Drubin lab (38). All other expression constructs generated for this study were made using Gibson Assembly (New England Biolabs) according to the manufacturer's instructions. All primers used in cloning are listed in Table 1. The lentiviral expression constructs pLVX-TagGFP2-Nck and pLVX-TagGFP2-N-WASP were generated by cloning a TagGFP2 fragment into ApaI/NotI sites of pLVX-GFP-Nck (66) and pLVX-GFP-N-WASP (31), respectively. The A36-TagGFP2 construct was made by assembling the fusion fragment and cloning it into NotI/HindIII sites of an A36 targeting vector (66). This vector was modified using mutagenic primers to obtain the A36(132F)-TagGFP2 construct. The A36ΔNPF-TagGFP2 construct was made by assembling a fusion fragment and cloning into SpeI/HindIII sites of the A36 targeting vector. The B5-TagGFP2 construct was made by cloning a TagGFP2 fragment into NotI/XbaI sites of a B5-YFP targeting vector (39). To generate the TagGFP2-F13 targeting vector, the F13 gene and 423 bp of upstream and downstream sequences were amplified from the WR strain of Vaccinia virus genomic DNA. This was assembled with a TagGFP2 fragment and cloned into NotI/HindIII sites of pBS SKII. To generate the TagGFP2-A33 targeting vector, the fusion construct flanked by 300 bp of genomic sequence upstream and downstream of the A33 coding region was obtained as a synthetic DNA fragment (IDT, gblock). This was cloned into NotI/HindIII sites of pBS SKII. SnapGene software (from Insightful Science; available at snapgene.com) was used to plan and visualize cloning strategies and to analyze sequencing results.

**TABLE 1** Primers

| No. | Sequence | Construct(s) |
|---|---|---|
| AB123 | TCGACGGTACCGCGGGCCCATGGGAGGCGAAGAGCTGTTTGCTG | pLVX-TagGFP2-Nck pLVX-TagGFP2-N-WASP |
| AB124 | TGCCATTCCTCCGCGGCCGCCTGTCCCGGAACCTGCCCT | pLVX-TagGFP2-Nck |
| AB125 | GCCGCCGCCGCGGCCGCCTGTCCCGGAACCTGCCCT | pLVX-TagGFP2-N-WASP |
| AB084 | GTCGTATCATTGGTGGGGTCGGGGTCGGATCCACCGGTCGCCACCGGAGGCGAAGAGCTGTTT | pBS-SKII-A36-TagGFP2, pBS-SKII-A36ΔNPF-TagGFP2 |
| AB085 | GTCGTATCATTGGTGGGGTCGGGGTCGGATCCACCGGTCGCCACCGGAGGCGAAGAGCTGTTT | pBS-SKII-A36-TagGFP2, pBS-SKII-A36ΔNPF-TagGFP2 |
| AB086 | GCAGGTTCCGGGACATAATTAGTTTCCTTTTTATAAAATTGAAG | pBS-SKII-A36-TagGFP2, pBS-SKII-A36ΔNPF-TagGFP2 |
| AB052 | GGTCGACGGTATCGATAAGCTTTATCTATAGAGATAACAC | pBS-SKII-A36-TagGFP2, pBS-SKII-A36ΔNPF-TagGFP2 |
| AB096 | AAACAATAAATATTGAACTAGTAGTACGTATATTGAGC | pBS-SKII-A36ΔNPF-TagGFP2 |
| AB081 | GGTGGCGACCGGTGGATCCGACCCCGACCCCACCAATGATACGACCGAT | pBS-SKII-A36ΔNPF-TagGFP2 |
| AB009 | CAGACTATTTTTCAGAACACTACAGTAGTA | pBS-SKII-A36(132F)-TagGFP2 |
| AB010 | CTGTAGTGTTCTGAAAAATAGTCTGT | pBS-SKII-A36(132F)-TagGFP2 |
| MHG055 | ATTGGAGCTCCACCGCGGTGGCGGCCGCCGCCATCGTCGGTGTGTTGTC | pBS-SKII-TagGFP2-F13 |
| AB135 | GCAAACAGCTCTTCGCCTCCCATTTAGTTAACATAAAAACTTA | pBS-SKII-TagGFP2-F13 |
| AB136 | TAAGTTTTTATGTTAACTAAATGGGAGGCGAAGAGCTGTTTGC | pBS-SKII-TagGFP2-F13 |
| AB137 | GTACCGATGCAAATGGCCACATGCGGCCGCCTGTCCCGGAACCTGCCCT | pBS-SKII-TagGFP2-F13 |
| AB131 | AGGGCAGGTTCCGGGACATAAATATAAATCCGTTAAAATAA | pBS-SKII-B5-TagGFP2 |
| AB132 | TCCCCGCGGGTGGGCGCTCTAGAACTAGTGGGATCCTATACCATT | pBS-SKII-B5-TagGFP2 |
| AB133 | ATTGCTACCGGGCGGCGCGGCCGCCGGAGGCGAAGAGCTGTTTGC | pBS-SKII-B5-TagGFP2 |
| AB134 | TTATTTTAACGGATTTATATTTATGTCCCGGAACCTGCCCT | pBS-SKII-B5-TagGFP2 |

## Cell lines

HeLa cells were maintained in minimal essential medium (MEM) and MEFs, and A549 cells were maintained in Dulbecco's Modified Eagle Medium. All cells were supplemented with 10% FBS (fetal bovine serum), 100 U/mL penicillin, and 100 µg/mL streptomycin at 37°C and 5% $CO_2$. The HeLa cell line stably expressing LifeAct-iRFP670 (37) was previously generated in the Way lab. Nck –/– MEFs (67) and N-WASP–/–MEFs (68) were provided by the late Tony Pawson (Samuel Lunenfeld Research Institute, Toronto, Canada) and Scott Snapper (Harvard Medical School, Boston, MA, USA), respectively. A549 cells were obtained from ATCC. For this study, lentiviral expression vectors were used to stably express TagGFP2-Nck in Nck–/– MEFs and TagGFP2-N-WASP in N-WASP–/–MEFs. All cell lines were generated using the lentivirus Trono group second generation packaging system (Addgene) and selected using puromycin resistance (2 µg/mL) as previously described (69). Expression of the relevant fusion proteins was confirmed by live imaging and immunoblot analysis (Fig. S4). The following primary antibodies were used: anti-Nck (BD transduction; 1:1,000), anti-vinculin (Sigma #V4505; 1:2,000), anti-N-WASP (Cell Signalling #4848S; 1:1,000), and anti-TagGFP2 antibody (Evrogen #AB121; 1:3,000). HRP-conjugated secondary antibodies were purchased from The Jackson Laboratory.

## Viral plaque assays

Plaque assays were performed in confluent BS-C-1 cell monolayers. Cells were infected with the relevant Vaccinia virus at a multiplicity of infection (MOI) = 0.1 in serum-free MEM for 1 hour. The inoculum was replaced with a semi-solid overlay consisting of a 1:1 mix of MEM and 2% carboxymethyl cellulose. Cells were fixed with 3% formaldehyde at 72 hours post-infection and subsequently visualized with crystal violet cell stain as previously described (34). To determine plaque size, the diameter of well-separated plaques was measured using the Fiji line tool (70).

## Construction of recombinant Vaccinia viruses

Recombinant viruses expressing RFP-A3 with untagged variants A36(132F) and A36ΔNPF were generated previously in the Way lab (35, 37). In this study, recombinant Vaccinia viruses expressing A36-TagGFP2, B5-TagGFP2, TagGFP2-A33, TagGFP2-F13, A36-(132F)-TagGFP2, and A36ΔNPF-TagGFP2 at their endogenous loci were generated in a parental WR viral strain expressing RFP-A3. Recombinants were isolated by selecting viral plaques positive for TagGFP2 fluorescence as described previously (66). To introduce relevant constructs into their genomic loci, HeLa cells infected with WR (RFP-A3) at MOI = 0.05 were transfected with the appropriate pBS SKII targeting vectors using Lipofectamine2000 (Invitrogen) as described by the manufacturer. When all cells displayed cytopathic effect at 48–72 hours post-infection, they were lysed, and serial dilutions of the lysates were used to infect confluent BS-C-1 cell monolayers in a plaque assay (see above). RFP and TagGFP2 positive plaques were picked, and plaque lysates were used to infect fresh BS-C-1 cell monolayers over at least three rounds of plaque purification. Successful recombination at the correct locus, loss of the parent variant, and virus purity were verified by PCR, sequencing, and immunoblot. The following primary antibodies were used: anti-vinculin (Sigma #V4505; 1:2,000), anti-TagGFP2 (Evrogen #AB121; 1:3,000), anti-N1 [(71); 1:3,000], anti-A36 [(20); 1:3,000], anti-B5 [(5); 1:5,000], anti-F13L-N [(13); 1:5,000], and anti-A33-RL2 [(20); 1:1,000]. HRP-conjugated secondary antibodies were purchased from The Jackson Laboratory. For all experiments, recombinant viruses were purified through a sucrose cushion before use and storage.

## Vaccinia virus infection for imaging

For live and fixed cell imaging, cells plated on fibronectin-coated MatTek dishes (MatTek corporation) or coverslips were infected with the relevant Vaccinia virus recombinant in

serum-free MEM at MOI = 1. After 1 hour at 37°C, the serum-free MEM was removed and replaced with complete MEM. Cells were incubated at 37°C until further processing.

## Transient transfection and drug treatment

All transfections prior to imaging were carried out using FUGENE (Promega) as described by the manufacturer. To visualize filamentous actin in live cells, Vaccinia-infected A549 cells and MEFs were transiently transfected with pE/L-LifeAct-iRFP670 1 hour after infection. Plasmids expressing nanocage constructs were transfected into HeLa cells or MEFs at least 16 hours prior to imaging along with a plasmid expressing cytosolic mCherry to easily identify transfected cells. To induce nanocage tethering to the plasma membrane, 500 nM AP21967 (Takara) was added to the media at least 30 minutes prior to imaging.

## Immunofluorescence

At 8 (HeLa) or 16 hours (MEFs) post-infection, cells were fixed with 4% paraformaldehyde in PBS for 10 minutes, blocked in cytoskeletal buffer (1 mM MES, 15 mM NaCl, 0.5 mM EGTA, 0.5 mM MgCl$_2$, and 0.5 mM glucose, pH 6.1) containing 2% (vol/vol) fetal calf serum and 1% (wt/vol) BSA (bovine serum albumin) for 30 minutes, and then permeabilized with 0.1% Triton-X/PBS for 5 minutes. To visualize CEV, cells were stained with a monoclonal antibody against B5 [19C2, rat, 1:1,000 (5)] followed by an Alexa Fluor 647 anti-rat secondary antibody (Invitrogen; 1:1,000 in blocking buffer) prior to permeabilization of the cells with detergent. Other primary antibodies used were anti-Nck (Millipore #06-288; 1:100) and anti-N-WASP (Cell Signalling #4848S; 1:100) followed by Alexa Fluor 568 conjugated secondary antibodies (Invitrogen; 1:1,000 in blocking buffer). Actin tails were labeled with Alexa Fluor 488 phalloidin (Invitrogen; 1:500). Coverslips were mounted on glass slides using Mowiol (Sigma). Coverslips were imaged on a Zeiss Axioplan2 microscope equipped with a 63×/1.4 NA Plan-Achromat objective and a Photometrics Cool Snap HQ cooled charge-coupled device camera. The microscope was controlled with MetaMorph 7.8.13.0 software.

## Live-cell imaging

Live-cell imaging experiments were performed starting at 8 or 16 hours post-infection in complete MEM (10% FBS) in a temperature-controlled chamber at 37°C. For all molecule-counting experiments, cells were imaged on a Zeiss Axio Observer spinning-disk microscope equipped with a Plan Achromat 100×/1.46 NA oil lens, an Evolve 512 camera, and a Yokogawa CSUX spinning disk. The microscope was controlled by the SlideBook software (3i Intelligent Imaging Innovations). To generate a calibration curve using TagGFP2 nanocages, image stacks of 10 Z-slices 0.1 µm apart were acquired using an exposure of 100 ms (unless stated otherwise) for 10 seconds at 1-second intervals. When quantifying TagGFP2 molecules at virus particles, a single Z stack was acquired followed by a single plane image of RFP (virus) and iRFP (LifeAct) channels to help identify virus particles generating actin tails. To determine virus speed, images were acquired using a 63×/1.4 Ph3 M27 oil lens in RFP (virus) and iRFP (LifeAct) channels for 50 seconds at 1 Hz.

## Image analysis and quantitation

For molecule counting analysis, average intensity Z projections of GFP channel images were generated in Fiji. The ComDet v.0.5.5 plugin for ImageJ (https://github.com/UU-cellbiology/ComDet) was used to identify fluorescent spots and measure their background-subtracted integrated intensity. Only tethered nanocages, which remained in the frame over 10 seconds of imaging, were used for analysis. Intensities obtained from 24-, 60-, 120-, and 180-mers imaged at 100-ms exposure were plotted as a function of predicted TagGFP2 number per nanocage. A line of linear fit through the origin was applied by linear least-squares fitting to generate the calibration curve used in this study. To calculate TagGFP2 numbers at Vaccinia CEV, fluorescent spots identified by the

ComDet v.0.5.5 plugin were overlaid with RFP (virus)/iRFP (actin) images to only select particles associated with actin tails.

To analyze virus motility, two-color time-lapse movies of HeLa cells stably expressing LifeAct-iRFP670 infected with the relevant recombinant virus labeled with RFP-A3 were used. The velocity of virus particles in the RFP channel was measured using a Fiji plugin developed by David Barry (the Francis Crick Institute) as previously described (66).

## Whole-cell lysate immunoblot

HeLa cells and MEFs infected with WR were lysed at 8 and 16 hours post-infection, respectively, in PBS containing 1% SDS, a cocktail of protease and phosphatase inhibitors [cOmplete (Roche), PHOSstop (Roche)] and Benzonase (Millipore). Proteins from these lysates were resolved on an SDS-PAGE, and the levels of total B5 and A36 were determined by immunoblot analysis using anti-A36 [(20); 1:3,000] and anti-B5 [(5); 1:5,000]. As loading controls, anti-vinculin (Sigma #V4505; 1:2,000) and anti-N1 [(71); 1:3,000] were used. HRP-conjugated secondary antibodies were purchased from The Jackson Laboratory.

## Statistical analysis and figure preparation

All data are presented as means $\pm$ SD For all experiments, means of at least three independent experiments (i.e., biological replicates) were used to determine statistical significance by a Welch's $t$-test (comparing only two conditions), Tukey's multiple comparison test (comparing multiple conditions with each other), or a Dunnett's multiple comparison test (comparing multiple conditions with a control). All data are represented as SuperPlots to allow assessment of the data distribution in individual experiments (72). SuperPlots were generated using the SuperPlotsOfData webapp (73) and GraphPad Prism 9. All data were analyzed using GraphPad Prism 9 or the Super-PlotsOfData webapp. Schematics were created with BioRender.com. Final figures were assembled using Keynote software.

## ACKNOWLEDGMENTS

We thank members of the Way Laboratory for useful discussions and suggestions. We thank David Drubin and Matthew Akamatsu (UC Berkeley) for sharing nanocage plasmids and helping optimize their imaging. We also thank David Barry (Francis Crick Institute) and Stephen J. Royle (University of Warwick) for image analysis advice, and Miguel Hernández González (Way Lab) for help in generating the TagGFP2-F13 expression construct. Special thanks to Jeremy Carlton and Snezhka Oliferenko (Francis Crick Institute) for their helpful comments on the manuscript.

M.W. was supported by the Francis Crick Institute, which receives its core funding from Cancer Research UK (CC2096), the UK Medical Research Council (CC2096), and the Wellcome Trust (CC2096).

For the purpose of Open Access, the authors have applied a CC by public copyright license to any author accepted manuscript version arising from this submission.

## AUTHOR AFFILIATIONS

[1]Cellular Signalling and Cytoskeletal Function Laboratory, The Francis Crick Institute, London, United Kingdom
[2]Department of Infectious Disease, Imperial College, London, United Kingdom

## AUTHOR ORCIDs

Angika Basant http://orcid.org/0000-0002-4754-6647
Michael Way http://orcid.org/0000-0001-7207-2722

## FUNDING

| Funder | Grant(s) | Author(s) |
|---|---|---|
| Cancer Research UK (CRUK) | CC2096 | Michael Way |
| UKRI \| Medical Research Council (MRC) | CC2096 | Michael Way |
| Wellcome Trust (WT) | CC2096 | Michael Way |

## AUTHOR CONTRIBUTIONS

Angika Basant, Conceptualization, Formal analysis, Investigation, Methodology, Visualization, Writing – original draft, Writing – review and editing | Michael Way, Conceptualization, Funding acquisition, Project administration, Resources, Supervision, Writing – original draft, Writing – review and editing

## ADDITIONAL FILES

The following material is available online.

### Supplemental Material

**Fig. S1 (Spectrum01529-23-s0001.tif).** Validation of TagGFP2-labeled recombinant viruses.
**Fig. S2 (Spectrum01529-23-s0002.tif).** Validation of TagGFP2-A33 recombinant virus.
**Fig. S3 (Spectrum01529-23-s0003.tif).** Validation of fluorescent nanocages.
**Fig. S4 (Spectrum01529-23-s0004.tif).** Validation of stable cell lines expressing TagGFP2-Nck and TagGFP2-N-WASP.
**Supplemental legends (Spectrum01529-23-s0005.docx).** Legends for Fig. S1 to S4.

### Open Peer Review

**PEER REVIEW HISTORY (review-history.pdf).** An accounting of the reviewer comments and feedback.

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
