## [Reviewer comments · Microbiology Spectrum]

Microbiology Spectrum

The level of Nck rather than N-WASP determines the rate of actin-based motility of Vaccinia

Angika Basant and Michael Way

Corresponding Author(s): Michael Way, The Francis Crick Institute

Review Timeline:

Submission Date:	April 11, 2023
Editorial Decision:	May 16, 2023
Revision Received:	August 24, 2023
Accepted:	September 3, 2023

Editor: Wen Chang

Reviewer(s): The reviewers have opted to remain anonymous.

Transaction Report:

DOI: <https://doi.org/10.1128/spectrum.01529-23>

May 16, 2023

Dr. Michael Way
The Francis Crick Institute
Cellular signalling and cytoskeletal function laboratory
1 Midland Road
London
United Kingdom

Re: Spectrum01529-23 (The level of Nck rather than N-WASP determines the rate of actin-based motility of Vaccinia)

Dear Dr. Michael Way:

Thank you for submitting the manuscript to Microbiology Spectrum. Your manuscript has been carefully reviewed by two experts in the field. Please revise your manuscript accordingly.

Link Not Available

Sincerely,

Wen Chang

Journals Department
Reviewer comments:

Reviewer #1 (Comments for the Author):

Orthopoxviruses such as vaccinia virus utilize actin polymerisation for the formation of actin tails to propel newly released virions away from infected cells. Actin tail formation has served as a useful paradigm for actin polymerization. In the manuscript by Basant and Way, they utilize this system to make careful calculations of the stoichiometry of the various proteins involved in actin tail formation. The manuscript is well-written and the conclusions justified. I feel that there are only a few small issues that need to be addressed.

1) It should be noted in the text that the addition of TagGFP2 to A36, B5 and F13 significantly reduces that amount of these

proteins that are expressed (Suppl1).

- 2) Wording is a bit tricky when referring to where the various proteins are localized. Technically, none of these are "on" CEV but lie underneath them.
- 3) The authors should explain how "inclusion of an actin marker ensures that only CEV are analyzed" (line 117).
- 4) Viruses A36-Y132 and A36 Δ NPF are not described in the text nor the materials and methods. Exactly how they were created should be described (or cited).
- 5) The sentence on line 228 is incomplete.
- 6) Looking at the diagram in Fig. 1, the assumption (line 246) that the area under the CEV (plasma membrane) is equal to the size of the outer IEV membrane is inaccurate. Once the IEV fuses with the outer membrane, much of the IEV membrane (lipids and proteins) would likely diffuse away and only the part that is directly under the CEV virion would contain proteins interacting with the CEV. This assumption is also made on line 293.

Reviewer #2 (Comments for the Author):

This manuscript by Basant and Way continues this group's longstanding dissection of actin-based motility in the Vaccinia virus system. Here, they rigorously quantify the number of A36, Nck, and N-WASP molecules recruited to cell-associated enveloped virus particles (CEV) associated with actin comets in infected cells. They do this both in human HeLa cells and mouse embryonic fibroblasts (MEF), which must be used to quantify Nck and N-WASP levels. Somewhat surprisingly, they find that A36 levels associated with CEV are higher in MEF cells than HeLa, perhaps due to the slower kinetics of infection in murine cells (allowing more time for A36 to accumulate). Quantification in MEF cells demonstrated that A36:Nck:N-WASP ratios were approximately 4:2:1 using wt virus. Mutation of the Grb2 SH2 binding site in A36 modestly decreased Nck binding, while mutation of the intersectin binding site more strongly decreased Nck binding; both mutations strongly decreased N-WASP binding to CEV. Finally, and perhaps most surprisingly, using these mutants the speed of actin motility on CEV was shown to be inversely proportional to the level of Nck binding.

Overall, the experiments are technically sound and carefully interpreted, and thus provide valuable new information for understanding the design principles that underlie localized actin polymerization. Solid data on numbers of molecules will likely be valuable in quantitative modeling studies as well.

Specific points:

1. One potential technical issue with the quantification of Nck and N-WASP is that it is impossible to measure more than one of the three molecules (A36, Nck, N-WASP) in the same cells--not only is only one tag used for quantification, but Nck and N-WASP must be quantified in knockout cells reconstituted with tagged Nck and N-WASP. This to some extent weakens the ability to correlate amounts of the three different molecules and their ratios. In principle, reconstitution with tagged Nck and N-WASP could affect other components associating with CEV. I am not sure this can be experimentally addressed, but the issue should at least be discussed briefly in the text.
2. Similarly, it would be useful to quantify the number of mutant A36 molecules (Y132F and deltaNPF) associated with CEV, to ensure that differential expression of the mutants is not what is driving differences observed in Nck and N-WASP recruitment.
3. A36-Nck-Grb2-NWASP complexes (perhaps also including WIP, intersectin, etc.) involve the type of relatively weak multivalent interactions that have been shown to lead to formation of biomolecular condensates. Indeed, the first in vitro studies of such condensates used constructs that mimic multivalent Nck/N-WASP interactions. It might be interesting to discuss potential effects of condensate formation on dwell-time of components, and how this might affect the extent of actin polymerization and its speed. I know this group has previously looked at apparent off-rates for various components in the Vaccinia system, which may be relevant.
4. The authors are careful to provide multiple validations for extrapolating outside the existing standard curve to quantify A36 levels (Fig. 2). However, the quantification of B5 and F13 are much further outside the standard curve. The authors should at minimum comment on the reasons they feel this extrapolation is justified.
5. P. 7, line 199: The statement of changes in ratios of A36 to Nck to N-WASP should perhaps be phrased differently; currently it says differences in NPF mutants are "significantly reduced" while differences in Y132F mutants are "reduced." If the intent is to discuss statistical significance, my guess is only Nck binding to the NPF mutant would be significant.
5. Discussion, p. 9: The authors speculate that for the deltaNPF mutant, the level of Nck binding is in a "sweet spot" to maximize activation of Arp2/3 by N-WASP. This argument would be strengthened by data from an A36 mutant that binds even less Nck, which might be expected to be less efficient/induce slower comet tails. Of course lack of Nck binding gives no comet tails, but another data point would be helpful here.
6. Do the authors know why the stoichiometry of Nck binding to A36 in CEV is roughly 0.5? Some potential explanations include

substoichiometric A36 phosphorylation, or the local concentration of Nck being near the Kd for binding to A36.

Staff Comments:

Preparing Revision Guidelines

Please return the manuscript within 60 days; if you cannot complete the modification within this time period, please contact me. If you do not wish to modify the manuscript and prefer to submit it to another journal, please notify me of your decision immediately so that the manuscript may be formally withdrawn from consideration by Microbiology Spectrum.

Response to reviewer comments:

Please note that the line numbers mentioned here correspond to the final text (without track changes marked-up).

Reviewer #1 (Comments for the Author):

Orthopoxviruses such as vaccinia virus utilize actin polymerisation for the formation of actin tails to propel newly released virions away from infected cells. Actin tail formation has served as a useful paradigm for actin polymerization. In the manuscript by Basant and Way, they utilize this system to make careful calculations of the stoichiometry of the various proteins involved in actin tail formation. The manuscript is well-written and the conclusions justified. I feel that there are only a few small issues that need to be addressed.

1) It should be noted in the text that the addition of TagGFP2 to A36, B5 and F13 significantly reduces that amount of these proteins that are expressed (Suppl1).

The following has been added in line 138 of the revised text “The overall expression levels of the tagged proteins in infected cells are reduced when compared to untagged controls (Fig S1).”

2) Wording is a bit tricky when referring to where the various proteins are localized. Technically, none of these are "on" CEV but lie underneath them.

We agree that it is challenging to appropriately describe the localisation of protein with respect to virus particles in each case. We have rephrased sentences where possible for more clarity, for eg. in line 158 of the revised text “Nck and N-WASP are essential for this process, but how many molecules of each are recruited beneath CEV on the plasma membrane?” and line 224 of the revised text “To test whether these differences are caused by varying levels of A36 associated with CEV, we generated recombinants expressing A36(132F)-TagGFP2 and A36 Δ NPF-TagGFP2 at the endogenous locus.”

3) The authors should explain how "inclusion of an actin marker ensures that only CEV are analyzed" (line 117).

We have now clarified this point in line 118 of the revised text which reads as follows “Co-localisation of A36-TagGFP2 and RFP-A3 signals occurs on IEV and CEV, but the inclusion of an actin marker ensures that only CEV are analysed, as actin tails are not generated by IEV in the cytoplasm of infected cells (Reitdorf et al., 2001 & Hollinshead et al., 2001).”

4) Viruses A36-Y132 and A36 Δ NPF are not described in the text nor the materials and methods. Exactly how they were created should be described (or cited).

We apologise for this omission in the original submission. These viruses were previously generated in the Way lab. The relevant papers are now cited in line 526 of the revised text “Recombinant viruses expressing RFP-A3 with untagged variants A36(132F) and A36 Δ NPF were generated previously in the Way lab (Weisswange et al., 2009 & Snetkov et al., 2016).”

5) The sentence on line 228 is incomplete.

We apologise for this omission in the original submission. This has now been corrected to read “Previous studies have taken advantage of the known structure of viruses to generate fluorescence calibration standards by tagging defined numbers of viral capsid components with fluorescent proteins” (line 246 in the revised text).

6) Looking at the diagram in Fig. 1, the assumption (line 246) that the area under the CEV (plasma membrane) is equal to the size of the outer IEV membrane is inaccurate. Once the IEV fuses with the

outer membrane, much of the IEV membrane (lipids and proteins) would likely diffuse away and only the part that is directly under the CEV virion would contain proteins interacting with the CEV. This assumption is also made on line 293.

We think that there is insufficient data available to ascertain the extent to which lipids and proteins diffuse away when IEV fuse with the plasma membrane. In line 308 of the revised text, we make clear the assumptions being made “Assuming that no membrane is lost, and proteins do not diffuse significantly when IEVs fuse with the plasma membrane, our data indicate that A36 is present at a density of 0.2-0.5 molecules per 100 nm² based on the outer surface of 405,037 nm² for IEV.”

Reviewer #2 (Comments for the Author):

This manuscript by Basant and Way continues this group's longstanding dissection of actin-based motility in the Vaccinia virus system. Here, they rigorously quantify the number of A36, Nck, and N-WASP molecules recruited to cell-associated enveloped virus particles (CEV) associated with actin comets in infected cells. They do this both in human HeLa cells and mouse embryonic fibroblasts (MEF), which must be used to quantify Nck and N-WASP levels. Somewhat surprisingly, they find that A36 levels associated with CEV are higher in MEF cells than HeLa, perhaps due to the slower kinetics of infection in murine cells (allowing more time for A36 to accumulate). Quantification in MEF cells demonstrated that A36:Nck:N-WASP ratios were approximately 4:2:1 using wt virus. Mutation of the Grb2 SH2 binding site in A36 modestly decreased Nck binding, while mutation of the intersectin binding site more strongly decreased Nck binding; both mutations strongly decreased N-WASP binding to CEV. Finally, and perhaps most surprisingly, using these mutants the speed of actin motility on CEV was shown to be inversely proportional to the level of Nck binding.

Overall, the experiments are technically sound and carefully interpreted, and thus provide valuable new information for understanding the design principles that underlie localized actin polymerization. Solid data on numbers of molecules will likely be valuable in quantitative modeling studies as well.

Specific points:

1. One potential technical issue with the quantification of Nck and N-WASP is that it is impossible to measure more than one of the three molecules (A36, Nck, N-WASP) in the same cells--not only is only one tag used for quantification, but Nck and N-WASP must be quantified in knockout cells reconstituted with tagged Nck and N-WASP. This to some extent weakens the ability to correlate amounts of the three different molecules and their ratios. In principle, reconstitution with tagged Nck and N-WASP could affect other components associating with CEV. I am not sure this can be experimentally addressed, but the issue should at least be discussed briefly in the text.

This is now discussed in line 198 of the revised text “This ratio is our best estimate, as unfortunately due to the nature of the fluorescent tag and lack of double Nck/N-WASP null cells, we cannot simultaneously measure the numbers of any pair of the three molecules in a single cell.”

2. Similarly, it would be useful to quantify the number of mutant A36 molecules (Y132F and deltaNPF) associated with CEV, to ensure that differential expression of the mutants is not what is driving differences observed in Nck and N-WASP recruitment.

To address this issue, we generated two new recombinant viruses where the A36 Y132F and A36 Δ NPF mutants were tagged with TagGFP2. Data shown in Fig 6C and line 226 of the revised manuscript demonstrate that the number of A36 molecules present at CEV inducing actin tails does not vary significantly across mutants.

3. A36-Nck-Grb2-NWASP complexes (perhaps also including WIP, intersectin, etc.) involve the type of

relatively weak multivalent interactions that have been shown to lead to formation of biomolecular condensates. Indeed, the first *in vitro* studies of such condensates used constructs that mimic multivalent Nck/N-WASP interactions. It might be interesting to discuss potential effects of condensate formation on dwell-time of components, and how this might affect the extent of actin polymerization and its speed. I know this group has previously looked at apparent off-rates for various components in the Vaccinia system, which may be relevant.

Our recent eLife paper (Basant & Way, 2022) suggests that spatial constraints and network organization are important parameters in determining signalling output. The role of condensates in Vaccinia actin tail formation remains to be investigated. Moreover, previous work on Nck/N-WASP interactions in condensate formation have used *in vitro* systems or situations that lack spatial constraints, which are clearly important for Vaccinia. Given this, we felt it was not appropriate to discuss the impact of condensates in the context of our results especially in the absence of any analysis of protein dynamics. We have, however, added "It will be interesting to investigate how these parameters influence the turnover of Nck and N-WASP beneath CEV (Weisswange et al., 2009)" (line 282) in the discussion of the revised text to bring up this issue to the reader.

4. The authors are careful to provide multiple validations for extrapolating outside the existing standard curve to quantify A36 levels (Fig. 2). However, the quantification of B5 and F13 are much further outside the standard curve. The authors should at minimum comment on the reasons they feel this extrapolation is justified.

As the images for B5 and F13 intensities were obtained in the linear response range of the camera i.e. without any saturating pixels, we think our estimates are reliable. A comment on this has been added in line 147 of the revised text.

5. P. 7, line 199: The statement of changes in ratios of A36 to Nck to N-WASP should perhaps be phrased differently; currently it says differences in NPF mutants are "significantly reduced" while differences in Y132F mutants are "reduced." If the intent is to discuss statistical significance, my guess is only Nck binding to the NPF mutant would be significant.

To address this, lines 209-211 of the revised text now read as follows "Our measurements show that mutation of Tyr132 to phenylalanine (loss of Grb2) reduced the level of Nck and N-WASP by ~24% and 44% respectively N-WASP (Fig 6A, B). However, in the absence of the A36 NPF motifs, Nck numbers were ~61% lower while N-WASP was reduced by ~36% (Fig 6A, B)."

5. Discussion, p. 9: The authors speculate that for the deltaNPF mutant, the level of Nck binding is in a "sweet spot" to maximize activation of Arp2/3 by N-WASP. This argument would be strengthened by data from an A36 mutant that binds even less Nck, which might be expected to be less efficient/induce slower comet tails. Of course lack of Nck binding gives no comet tails, but another data point would be helpful here.

We agree that a fourth datapoint would indeed strengthen our argument. We tested other A36 mutants as candidates, but we were unable to find a condition where Nck levels at CEV particles were lower than those observed in the A36 Δ NPF virus.

6. Do the authors know why the stoichiometry of Nck binding to A36 in CEV is roughly 0.5? Some potential explanations include substoichiometric A36 phosphorylation, or the local concentration of Nck being near the K_d for binding to A36.

We currently do not have evidence on the stoichiometry of A36 phosphorylation, or the local concentration of Nck at virus particles (K_d for Nck1 binding to the A36 phosphopeptide *in vitro* has been calculated to be ~50 μ M (Frese et al., JBC 2006)).

September 3, 2023

Dr. Michael Way
The Francis Crick Institute
Cellular signalling and cytoskeletal function laboratory
1 Midland Road
London
United Kingdom

Re: Spectrum01529-23R1 (The level of Nck rather than N-WASP determines the rate of actin-based motility of Vaccinia)

Dear Dr. Michael Way:

Your manuscript has been accepted, and I am forwarding it to the ASM Journals Department for publication. You will be notified when your proofs are ready to be viewed.

One reviewer suggested that it is more accurate to state that " the amount of Nck correlates with velocity." so please modify accordingly.

Sincerely,

Wen Chang
Editor, Microbiology Spectrum
